# Modeling All-Atom Glycan Structures via Hierarchical Message Passing and Multi-Scale Pre-training

**Minghao Xu** [1 2 3]   **Jiaze Song** [1]   **Keming Wu** [4]   **Xiangxin Zhou** [5]   **Bin Cui** [1]   **Wentao Zhang** [1 3]

## Abstract

Understanding the various properties of glycans with machine learning has shown some preliminary promise. However, previous methods mainly focused on modeling the backbone structure of glycans as graphs of monosaccharides (*i.e.*, sugar units), while they neglected the atomic structures underlying each monosaccharide, which are actually important indicators of glycan properties. We fill this blank by introducing the **GlycanAA** model for **A**ll-**A**tom-wise **Glycan** modeling. GlycanAA models a glycan as a heterogeneous graph with monosaccharide nodes representing its global backbone structure and atom nodes representing its local atomic-level structures. Based on such a graph, GlycanAA performs *hierarchical message passing* to capture from local atomic-level interactions to global monosaccharide-level interactions. To further enhance model capability, we pre-train GlycanAA on a high-quality unlabeled glycan dataset, deriving the **PreGlycanAA** model. We design a *multi-scale mask prediction* algorithm to endow the model about different levels of dependencies in a glycan. Extensive benchmark results show the superiority of GlycanAA over existing glycan encoders and verify the further improvements achieved by PreGlycanAA. We maintain all resources at https://github.com/kasawa1234/GlycanAA.

## 1. Introduction

Glycans, complex macromolecules composed of sugar molecules, play pivotal roles in life science. They serve as essential structural components in cells, forming the backbone of extracellular matrices and cell membranes (Yanagishita, 1993). Based on such structures, they modulate intercellular communication (Liu & Wang, 2023) and impact biological processes such as immune response (Zhang, 2006) and cell differentiation (Lau et al., 2007). With the accumulation of glycan data in public repositories (Tiemeyer et al., 2017; Yamada et al., 2020), it is a promising way to understand various glycan properties and functions with data-driven methods like machine learning.

In this research direction, most existing works (Burkholz et al., 2021; Lundstrøm et al., 2022; Carpenter et al., 2022; Alkuhlani et al., 2023) model a glycan as a graph with monosaccharides (*i.e.*, sugar units) as its nodes, and use graph neural networks (GNNs) to predict various glycan properties, *e.g.*, glycosylation, immunogenicity, binding affinity with a protein, *etc.* Though performing well on some tasks, these methods fail to capture the atomic-level structures underlying each monosaccharide, which are actually important determinants of many glycan properties and functions. For example, atomic-level interactions between a glycan and a protein determine their binding affinity.

There have been some preliminary attempts at modeling all-atom-wise glycan structures with state-of-the-art small molecule encoders (Xu et al., 2024). However, because of the gap between a small molecule with tens of atoms and a glycan with hundreds of atoms (*i.e.*, essentially a macromolecule), these small molecule encoders are shown to be ineffective, which perform even worse than the models utilizing only monosaccharide-level information. Therefore, it is still to be answered how to realize the potential of all-atom glycan modeling on boosting glycan understanding.

To answer this question, in this work, we propose the **GlycanAA** model for **A**ll-**A**tom-wise **Glycan** modeling. Note that, a glycan naturally possesses a hierarchical structure with (1) atoms making up the local structure of each monosaccharide and (2) different monosaccharides making up the global backbone structure of the glycan. Inspired by this fact, we design GlycanAA based on a hierarchical modeling approach. Specifically, GlycanAA first represents a glycan as a heterogeneous graph consisting of (1) a set of atom nodes for its local structures and (2) a set of monosac-

---
*Equal contribution   [1]Peking University   [2]BioGeometry   [3]Zhongguancun Academy   [4]Tsinghua University   [5]University of Chinese Academy of Sciences. Correspondence to: Wentao Zhang <wentao.zhang@pku.edu.cn>.

*Proceedings of the 42^{nd} International Conference on Machine Learning*, Vancouver, Canada. PMLR 267, 2025. Copyright 2025 by the author(s).

charide nodes for its global structure. GlycanAA then performs *hierarchical message passing* to model from local atomic-level interactions to global monosaccharide-level interactions. In this way, GlycanAA can completely capture the covalent bonds forming each monosaccharide and the glycosidic bonds forming the whole glycan.

To further enhance the representation power of GlycanAA, we seek to endow it with the knowledge stored in abundant unlabeled glycan data. We resort to self-supervised pre-training to achieve this goal, where the **PreGlycanAA** model is developed as a pre-trained version of GlycanAA. Specifically, we first curate an unlabeled glycan dataset by selecting 40,781 high-quality glycan data from the GlyTouCan database (Tiemeyer et al., 2017). GlycanAA is then pre-trained on this dataset with a *multi-scale mask prediction* algorithm. In this algorithm, partial atom and monosaccharide nodes are masked at the input, and the model is asked to recover these masked nodes. Through this approach, the derived PreGlycanAA model acquires the dependencies between different atoms and monosaccharides in a glycan, leading to informative glycan representations.

We evaluate the proposed models on the GlycanML benchmark (Xu et al., 2024). Experimental results show that PreGlycanAA and GlycanAA respectively rank first and second on the benchmark, and they substantially outperform SOTA atomic-level small molecule encoders and glycan-specific monosaccharide-level encoders. We further demonstrate the effectiveness of the proposed hierarchical message passing and multi-scale mask prediction methods through extensive ablation studies.

## 2. Related Work

**Glycan modeling with machine learning.** With the growing size of experimental glycomics datasets, machine learning techniques are becoming increasingly important in glycoinformatics (Bojar & Lisacek, 2022; Li et al., 2022). Traditional machine learning approaches, such as support vector machines (SVMs), have been employed to learn patterns from mass spectrometry data (Kumozaki et al., 2015; Liang et al., 2014), predict glycosylation sites (Caragea et al., 2007; Li et al., 2015; Taherzadeh et al., 2019; Pitti et al., 2019), and classify glycans (Yamanishi et al., 2007). Alongside the advancements in deep learning, recent models have showcased the potential of deep learning in addressing glycomics challenges. Both sequence-based models (Bojar et al., 2020b;a; Pakhrin et al., 2021; Dai et al., 2021) and graph neural networks (GNNs) are utilized to predict various glycan properties on the datasets like N-GlyDE (Pitti et al., 2019) and SugarBase (Bojar et al., 2020b). Among all, GlycanML (Xu et al., 2024) established a comprehensive benchmark evaluating sequence-based models and GNNs on a diverse set of 11 tasks.

While GNNs have demonstrated their strong performance on specific tasks (Xu et al., 2024), their potential remains constrained by the underutilization of atomic-level information. Moreover, atomic-level encoders originally designed for small molecules have been shown to be ineffective in glycan modeling (Xu et al., 2024). In this study, we tackle these limitations by proposing the GlycanAA model, a hierarchical encoder for heterogeneous all-atom glycan graphs.

**Self-Supervised Pre-training (SSP) in the biological domain.** SSP has emerged as a powerful approach in deep learning, greatly improving the ability to learn informative and transferable representations from large-scale unlabeled data (Devlin, 2018; He et al., 2020).

In recent years, SSP has also gained remarkable success in the biological domain, where the availability of large-scale biological datasets makes pre-training techniques well-suited. For small molecules, SSP has improved molecular representations, facilitating tasks like molecular property prediction and drug discovery (Hu et al., 2019; Xia et al., 2022). Protein modeling is similarly benefited, with methods like protein language modeling (Elnaggar et al., 2021; Rives et al., 2021; Lin et al., 2022; Hayes et al., 2024), geometric structure pre-training (Zhang et al., 2023b; 2024) and multimodal approaches (Xu et al., 2023; Duy Nguyen & Son Hy, 2024). SSP also benefits DNA and RNA research with representative pre-trained models like DNABERT (Ji et al., 2021), DNAGPT (Zhang et al., 2023a), GenerRNA (Zhao et al., 2024), UNI-RNA (Wang et al., 2023b) and Evo (Nguyen et al., 2024).

Despite these advances, the potential of SSP in glycan modeling remains largely unexplored, presenting a new area of opportunity. In this work, we fill this gap by introducing the PreGlycanAA model which performs multi-scale pre-training on a high-quality unlabeled glycan dataset, leading to performance gains on various downstream glycan understanding tasks.

## 3. GlycanAA: All-Atom Glycan Modeling with Hierarchical Message Passing

We propose the GlycanAA model for all-atom-wise glycan modeling. In the following parts, we introduce its data representation method in Section 3.1 and its encoding approach in Section 3.2.

### 3.1. Heterogeneous Graph Representation of All-Atom Glycan Structure

For a glycan $g$, we represent its atomic-level structure as a heterogeneous graph $g = (\mathcal{V}_a, \mathcal{V}_m, \mathcal{E})$ composed of an atom node set $\mathcal{V}_a$, a monosaccharide node set $\mathcal{V}_m$ and an edge set $\mathcal{E}$, as graphically illustrated in Figure 1(a). We state the details of each graph component as below:

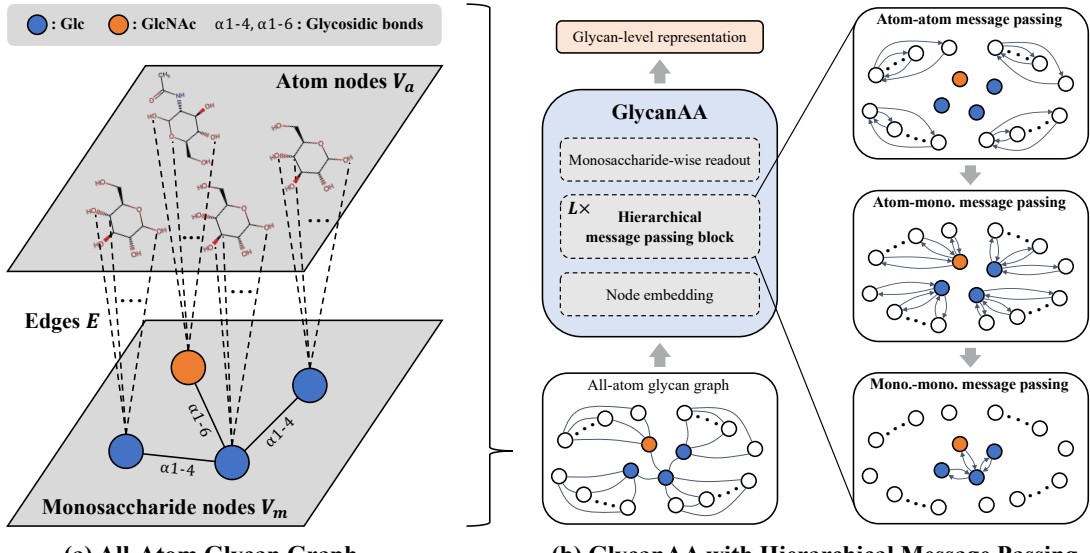

**(a) All-Atom Glycan Graph**  **(b) GlycanAA with Hierarchical Message Passing**

Figure 1: *Illustration of GlycanAA.* (a) GlycanAA represents a glycan as an all-atom heterogeneous graph with atom nodes, monosaccharide nodes and different types of edges between these nodes. (b) Based on such a graph, GlycanAA models atom-atom, atom-monosaccharide and monosaccharide-monosaccharide interactions through hierarchical message passing. *Abbr.*, Glc: Glucose, GlcNAc: N-Acetylglucosamine, mono.: monosaccharide.

- **Atom node set** $\mathcal{V}_a$: This node set contains all heavy atoms (*i.e.*, non-hydrogen atoms) in a glycan, *i.e.*, $\mathcal{V}_a = \{a_i\}_{i=1}^{N}$ ($a_i$ stands for an atom; $N$ denotes the number of atoms in glycan $g$).

- **Monosaccharide node set** $\mathcal{V}_m$: To clearly represent the backbone structure of a glycan, we further introduce a set of nodes representing all monosaccharides that make up the glycan, *i.e.*, $\mathcal{V}_m = \{m_j\}_{j=1}^{M}$ ($m_j$ stands for a monosaccharide; $M$ denotes the number of monosaccharides in glycan $g$).

- **Edge set** $\mathcal{E}$: We consider three kinds of edges to comprehensively represent atom-atom, atom-monosaccharide and monosaccharide-monosaccharide interactions, *i.e.*, $\mathcal{E} = \mathcal{E}_{aa} \cup \mathcal{E}_{am} \cup \mathcal{E}_{mm}$, as detailed below:

  - *Atom-atom edge set* $\mathcal{E}_{aa}$: This set of edges represent the atomic-level structure of each monosaccharide. Specifically, the covalent bonds in each monosaccharide are collected, and each bond along with its bond type (single, double, triple or aromatic bond) makes up an edge, *i.e.*, $\mathcal{E}_{aa} = \{(a, a', r) | r \in \{\text{single}, \text{double}, \text{triple}, \text{aromatic}\}\}$, where $(a, a', r)$ denotes an edge connecting atom $a$ to atom $a'$ with bond type $r$. We include both directions of a bond in this edge set.

  - *Atom-monosaccharide edge set* $\mathcal{E}_{am}$: We connect each atom with its corresponding monosaccharide, such that a monosaccharide is aware of its atomic-level information, and each atom recognizes the gly-

can backbone structure. This edge set is represented as $\mathcal{E}_{am} = \{(a, m, r_{am})\} \cup \{(m, a, r_{am})\}$, where each corresponding pair of atom $a$ and monosaccharide $m$ are connected by a bidirectional edge with the edge type $r_{am}$ indicating atom-monosaccharide interaction.

  - *Monosaccharide-monosaccharide edge set* $\mathcal{E}_{mm}$: We collect all glycosidic bonds in a glycan to represent its backbone structure. In specific, this edge set can be represented as $\mathcal{E}_{mm} = \{(m, m', r) | r \in \mathcal{R}_g\}$, where $(m, m', r)$ denotes an edge connecting monosaccharide $m$ to monosaccharide $m'$ with bond type $r$, and $\mathcal{R}_g$ denotes all possible types of glycosidic bonds, *e.g.*, alpha-1,6-glycosidic bond, beta-1,4-glycosidic bond, *etc.* We follow Thomès et al. (2021) to construct $\mathcal{R}_g$ and include both directions of a bond in this edge set.

### 3.2. Hierarchical Message Passing on All-Atom Glycan Graph

Based on the all-atom glycan graph introduced above, GlycanAA extracts glycan representations using the carefully-designed modules below. A graphical illustration is shown in Figure 1(b).

**Node embedding**: We employ two codebooks to store the embeddings of all possible types of atoms and monosaccharides, respectively. For each node, we look up the corresponding codebook to assign it an initial feature embedding.

**Hierarchical message passing**: A glycan possesses a hierarchical structure, where its local structure in each monosaccharide is formed by atoms and covalent bonds in between, and different monosaccharides are further connected by glycosidic bonds, deriving its global backbone structure. We propose to encode such a structure from local to global hierarchically, which is proven to be effective in modeling other biomolecules like small molecules (Yu & Gao, 2022; Han et al., 2023) and proteins (Hermosilla et al., 2020; Wang et al., 2023a). Specifically, in each message passing block, we sequentially perform atom-atom, atom-monosaccharide and monosaccharide-monosaccharide message passing to capture from local interactions to global interactions.

Note that, these interactions are essentially *multi-relational*, where atoms and monosaccharides interact with different types of covalent and glycosidic bonds. To fully model such interactions, we adopt relational graph convolution (RGConv) (Schlichtkrull et al., 2018) as the basic message passing module. Given a graph $g_0 = (\mathcal{V}_0, \mathcal{E}_0, \mathcal{R}_0)$ with node set $\mathcal{V}_0$, edge set $\mathcal{E}_0$ and relation (*i.e.*, edge type) set $\mathcal{R}_0$, RGConv updates node representations $Z_0 = \{z_i\}_{i=1}^{|\mathcal{V}_0|}$ by aggregating neighborhood information with per-relation convolutional operations:

$$Z_0' = \{z_i'\}_{i=1}^{|\mathcal{V}_0|} = \text{RGConv}(Z_0; \mathcal{V}_0, \mathcal{E}_0, \mathcal{R}_0),$$

$$z_i' = W_{\text{self}}\, z_i + \sigma\left(\text{BN}\left(\sum_{r \in \mathcal{R}_0} \sum_{v_j \in \mathcal{N}_r(v_i)} \frac{1}{|\mathcal{N}_r(v_i)|} W_r z_j\right)\right),$$

$$(1)$$

where $Z_0'$ denotes the updated node representations, $\mathcal{N}_r(v_i) = \{v_j | (v_j, v_i, r) \in \mathcal{E}_0\}$ are the neighbors of node $v_i$ with relation $r$, $W_r$ denotes the convolutional kernel matrix for relation $r$, and $W_{\text{self}}$ is the weight matrix for self-update. Here BN denotes a batch normalization layer, and we use a ReLU function as the activation $\sigma(\cdot)$.

Based on RGConv, we perform hierarchical message passing in three steps as below:

*Atom-atom message passing*:
$$Z_a' = \text{RGConv}(Z_a; \mathcal{V}_a, \mathcal{E}_{aa}, \mathcal{R}_{aa}), \quad (2)$$

*Atom-mono. message passing*:
$$(Z_a'', Z_m') = \text{RGConv}\big((Z_a', Z_m); \mathcal{V}_a \cup \mathcal{V}_m, \mathcal{E}_{am}, \mathcal{R}_{am}\big), \quad (3)$$

*Mono.-mono. message passing*:
$$Z_m'' = \text{RGConv}(Z_m'; \mathcal{V}_m, \mathcal{E}_{mm}, \mathcal{R}_{mm}), \quad (4)$$

where $\mathcal{R}_{aa}$ contains all types of covalent bonds, $\mathcal{R}_{am}$ stores the relation of atom-monosaccharide interaction, $\mathcal{R}_{mm}$ contains all types of glycosidic bonds, and "mono." is the abbreviation of monosaccharide. In this hierarchical process, atom representations $Z_a$ are first updated to $Z_a'$ by atom-atom message passing; atom and monosaccharide representations are then updated to $Z_a''$ and $Z_m'$ via atom-monosaccharide message passing; finally, monosaccharide

representations are updated to $Z_m''$ by monosaccharide-monosaccharide message passing.

**Monosaccharide-wise readout**: After $L$ blocks of hierarchical message passing, we get the final atom representations $Z_a^L$ and monosaccharide representations $Z_m^L$. We readout all monosaccharide nodes to get a glycan-level representation: $z_g = [\text{mean}(Z_m^L), \text{max}(Z_m^L)]$, where $\text{mean}(\cdot)$ and $\text{max}(\cdot)$ denote mean and max pooling, respectively, and $[\cdot, \cdot]$ stands for concatenation. We exclude atom nodes in the readout, considering that (1) many monosaccharides share similar or even the same atomic structure, leading to duplicating information in representation readout, and (2) useful atomic information has already been passed to monosaccharide nodes during atom-monosaccharide message passing. The ablation study in Section 5.3 also supports the superiority of monosaccharide-wise readout over all-node readout.

## 4. PreGlycanAA: Pre-train All-Atom Glycan Representations with Multi-Scale Mask Prediction

To further improve the representation power of GlycanAA, we endow it with the knowledge stored in abundant unlabeled glycan data through self-supervised pre-training, deriving the PreGlycanAA model. In the following parts, we introduce the setup of the pre-training dataset in Section 4.1 and the multi-scale pre-training algorithm in Section 4.2.

### 4.1. Curation of High-quality Unlabeled Glycan Dataset

To ensure the quality of pre-trained model, we aim to collect as much informative and clean glycan data as possible. We choose the GlyTouCan database (Tiemeyer et al., 2017) as the data source for its high recognition in the glycoscience domain and instant update of the latest glycan structures. We first collect all the glycans deposited in GlyTouCan, summing up to 219,857 glycans. Data cleaning is then performed based on the following criteria:

- **Data quality**: We discard all the glycans whose structures are not fully solved. In specific, if there is any monosaccharide or glycosidic bond with an undetermined type in a glycan, we regard it as a low-quality sample and remove it from pre-training.

- **Data integrity**: We preserve the glycan structures with a single connected component. Those samples with multiple components are discarded, so as to focus on learning the interactions within a single glycan structure.

- **Without data leakage**: We remove the glycans that occur in the dataset of any downstream task used in our experiments, so as to prevent data leakage during pre-training.

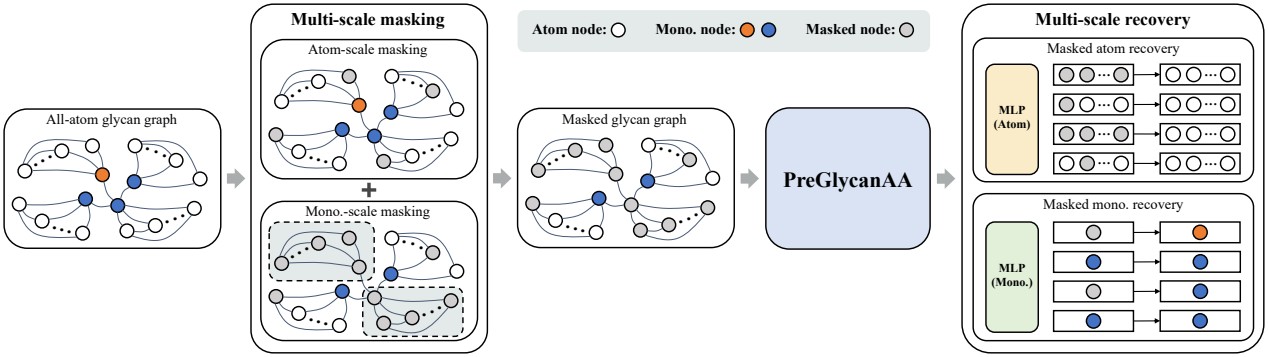

Figure 2: *Illustration of PreGlycanAA.* Upon an all-atom glycan graph, multi-scale masking derives a masked glycan graph with partially masked atoms and monosaccharides; PreGlycanAA learns multi-scale recovery to recover the complete glycan graph. *Abbr.*, mono.: monosaccharide.

After such a filtering process, we preserve a set of 40,781 high-quality, integral and data-leakage-proof glycan samples for pre-training.

### 4.2. Self-Supervised Pre-training via Multi-Scale Mask Prediction

To acquire the rich information underlying the curated unlabeled glycan dataset, we propose the PreGlycanAA model that pre-trains GlycanAA with a multi-scale mask prediction task, as illustrated in Figure 2. This algorithm endows the model with knowledge about the dependencies between different atoms and monosaccharides in a glycan, realized by the following schemes.

**Multi-scale masking**: During pre-training, it is desired to simultaneously acquire atom-atom, atom-monosaccharide and monosaccharide-monosaccharide dependencies. To achieve this goal, in an all-atom glycan graph (Section 3.1), we mask partial atom nodes and partial monosaccharide nodes, and the model is asked to recover these masked nodes by leveraging their neighboring atoms and monosaccharides. The two-scale masking is performed as below:

- *Atom-scale masking*: For all heavy atoms in a glycan, we randomly select a part of them with the ratio $\rho_a$, and they are represented by a type of `Unknown-Atom`.

- *Monosaccharide-scale masking*: We select partial monosaccharides in a glycan with the ratio $\rho_m$. On one hand, their corresponding monosaccharide nodes in the graph are masked with a type of `Unknown-Monosaccharide`. On other hand, to avoid the trivial prediction of a masked monosaccharide based on some of its characteristic atoms, we further mask all atom nodes corresponding to the selected monosaccharides with the `Unknown-Atom` type.

**Multi-scale recovery**: The PreGlycanAA model learns to recover all these masked nodes. Specifically, for a masked glycan graph $\tilde{g}$, the model first extracts its atom and monosaccharide representations $\widetilde{Z}_a = \{\tilde{z}_a | a \in \mathcal{V}_a\}$ and $\widetilde{Z}_m = \{\tilde{z}_m | m \in \mathcal{V}_m\}$ through hierarchical message passing. Based on such representations with rich neighborhood information, two MLP predictors $F_a$ and $F_m$ are respectively employed to recover masked atoms and monosaccharides, deriving the following pre-training loss:

$$\mathcal{L}_{\text{pretrain}} = \frac{1}{|\mathcal{V}_a^{\text{mask}}| + |\mathcal{V}_m^{\text{mask}}|} \left( \sum_{a \in \mathcal{V}_a^{\text{mask}}} \mathcal{L}_{\text{CE}}\big(F_a(\tilde{z}_a), y_a\big) + \sum_{m \in \mathcal{V}_m^{\text{mask}}} \mathcal{L}_{\text{CE}}\big(F_m(\tilde{z}_m), y_m\big) \right),$$

(5)

where $\mathcal{V}_a^{\text{mask}}$ and $\mathcal{V}_m^{\text{mask}}$ denote the set of masked atom nodes and masked monosaccharide nodes, $y_a$ and $y_m$ represent the ground-truth type of a masked atom node $a$ and a masked monosaccharide node $m$, and $L_{\text{CE}}$ stands for the cross-entropy loss. In summary, this pre-training method encourages the model to capture different levels of dependencies in a glycan by solving a glycan recovery problem.

## 5. Experiments

### 5.1. Experimental Setups

**Benchmark tasks**: We evaluate the effectiveness of the proposed models on the GlycanML benchmark (Xu et al., 2024). This benchmark contains a comprehensive set of 11 glycan property and function prediction tasks. Readers are referred to the original paper for detailed task descriptions and dataset statistics.

**Model setups**: For the sake of fair comparison with other baseline models in the GlycanML benchmark, both GlycanAA and PreGlycanAA are equipped with 3 hierarchical message passing blocks. For pre-training and downstream task training, we implement each prediction head as a

Table 1: Benchmark results on GlycanML. We report *mean (std)* for each experiment. The best, second-best, and third-best performances are denoted by **bold**, underline, and *italic*, respectively. *Abbr.*, Immuno.: Immunogenicity; Glycos.: Glycosylation; GlycanAA-SP: GlycanAA with a single message passing in each block; GlycanAA-AN: GlycanAA with all-node readout.

| Model | Taxonomy prediction | | | | | | | | Immuno. (*AUPRC*) | Glycos. (*Macro-F1*) | Interaction (*Spearman's* $\rho$) | Weighted Mean Rank |
|---|---|---|---|---|---|---|---|---|---|---|---|---|
| | Domain (*Macro-F1*) | Kingdom (*Macro-F1*) | Phylum (*Macro-F1*) | Class (*Macro-F1*) | Order (*Macro-F1*) | Family (*Macro-F1*) | Genus (*Macro-F1*) | Species (*Macro-F1*) | | | | |
| **Monosaccharide-level Glycan Sequence Encoders** | | | | | | | | | | | | |
| Transformer | $0.612_{(0.009)}$ | $0.546_{(0.079)}$ | $0.316_{(0.014)}$ | $0.235_{(0.022)}$ | $0.147_{(0.007)}$ | $0.114_{(0.039)}$ | $0.065_{(0.001)}$ | $0.047_{(0.008)}$ | $0.856_{(0.012)}$ | $0.729_{(0.069)}$ | $0.244_{(0.009)}$ | 16.09 |
| Shallow CNN | $0.629_{(0.005)}$ | $0.559_{(0.024)}$ | $0.388_{(0.024)}$ | $0.342_{(0.020)}$ | $0.238_{(0.016)}$ | $0.200_{(0.014)}$ | $0.149_{(0.009)}$ | $0.115_{(0.008)}$ | $0.776_{(0.027)}$ | $0.898_{(0.009)}$ | $0.261_{(0.008)}$ | 12.53 |
| LSTM | $0.621_{(0.012)}$ | $0.566_{(0.076)}$ | $0.413_{(0.036)}$ | $0.272_{(0.029)}$ | $0.174_{(0.023)}$ | $0.145_{(0.012)}$ | $0.098_{(0.016)}$ | $0.078_{(0.008)}$ | $0.912_{(0.068)}$ | $0.862_{(0.016)}$ | $0.280_{(0.001)}$ | 11.00 |
| ResNet | $0.635_{(0.009)}$ | $0.505_{(0.025)}$ | $0.331_{(0.061)}$ | $0.301_{(0.010)}$ | $0.183_{(0.082)}$ | $0.165_{(0.019)}$ | $0.112_{(0.018)}$ | $0.073_{(0.007)}$ | $0.754_{(0.124)}$ | $0.919_{(0.004)}$ | $0.273_{(0.004)}$ | 12.09 |
| **Monosaccharide-level Glycan Graph Encoders** | | | | | | | | | | | | |
| MPNN | $0.632_{(0.007)}$ | $0.638_{(0.050)}$ | $0.372_{(0.019)}$ | $0.326_{(0.015)}$ | $0.235_{(0.046)}$ | $0.161_{(0.004)}$ | $0.136_{(0.008)}$ | $0.104_{(0.009)}$ | $0.674_{(0.119)}$ | $0.910_{(0.006)}$ | $0.217_{(0.002)}$ | 18.34 |
| GCN | $0.635_{(0.001)}$ | $0.527_{(0.006)}$ | $0.325_{(0.024)}$ | $0.237_{(0.009)}$ | $0.147_{(0.005)}$ | $0.112_{(0.010)}$ | $0.095_{(0.009)}$ | $0.080_{(0.006)}$ | $0.688_{(0.023)}$ | $0.914_{(0.011)}$ | $0.233_{(0.009)}$ | 18.38 |
| GAT | $0.636_{(0.003)}$ | $0.523_{(0.007)}$ | $0.301_{(0.014)}$ | $0.265_{(0.012)}$ | $0.190_{(0.009)}$ | $0.130_{(0.005)}$ | $0.125_{(0.010)}$ | $0.103_{(0.009)}$ | $0.685_{(0.053)}$ | $0.934_{(0.038)}$ | $0.229_{(0.002)}$ | 16.94 |
| GIN | $0.632_{(0.004)}$ | $0.525_{(0.007)}$ | $0.322_{(0.046)}$ | $0.300_{(0.027)}$ | $0.179_{(0.002)}$ | $0.152_{(0.005)}$ | $0.116_{(0.022)}$ | $0.105_{(0.011)}$ | $0.716_{(0.051)}$ | $0.924_{(0.013)}$ | $0.249_{(0.004)}$ | 15.06 |
| CompGCN | $0.629_{(0.004)}$ | $0.568_{(0.047)}$ | $0.410_{(0.013)}$ | $0.381_{(0.024)}$ | $0.226_{(0.011)}$ | $0.193_{(0.012)}$ | $0.166_{(0.009)}$ | $0.138_{(0.014)}$ | $0.692_{(0.006)}$ | $0.945_{(0.002)}$ | $0.257_{(0.004)}$ | 12.19 |
| RGCN | $0.633_{(0.001)}$ | $0.647_{(0.054)}$ | *$0.462_{(0.033)}$* | $0.373_{(0.036)}$ | $0.251_{(0.012)}$ | $0.203_{(0.008)}$ | $0.164_{(0.003)}$ | $0.146_{(0.004)}$ | $0.780_{(0.006)}$ | $0.948_{(0.004)}$ | $0.262_{(0.005)}$ | 6.78 |
| PreRGCN | $0.636_{(0.005)}$ | $0.664_{(0.032)}$ | $0.451_{(0.023)}$ | $0.389_{(0.008)}$ | $0.265_{(0.015)}$ | $0.205_{(0.006)}$ | $0.172_{(0.010)}$ | $0.139_{(0.008)}$ | $0.781_{(0.019)}$ | $0.949_{(0.015)}$ | $0.263_{(0.018)}$ | *5.34* |
| GearNet | $0.471_{(0.005)}$ | $0.577_{(0.036)}$ | $0.395_{(0.025)}$ | $0.389_{(0.010)}$ | $0.256_{(0.007)}$ | $0.189_{(0.004)}$ | $0.165_{(0.003)}$ | $0.136_{(0.003)}$ | $0.740_{(0.015)}$ | $0.892_{(0.027)}$ | $0.248_{(0.004)}$ | 15.66 |
| GearNet-Edge | $0.628_{(0.009)}$ | $0.573_{(0.030)}$ | $0.396_{(0.010)}$ | $0.384_{(0.010)}$ | $0.262_{(0.006)}$ | $0.200_{(0.010)}$ | $0.177_{(0.008)}$ | $0.140_{(0.005)}$ | $0.768_{(0.023)}$ | $0.909_{(0.010)}$ | $0.250_{(0.003)}$ | 12.25 |
| ProNet | $0.627_{(0.007)}$ | $0.590_{(0.015)}$ | $0.438_{(0.012)}$ | $0.380_{(0.008)}$ | $0.242_{(0.005)}$ | $0.192_{(0.018)}$ | $0.146_{(0.010)}$ | $0.128_{(0.004)}$ | $0.778_{(0.019)}$ | $0.930_{(0.015)}$ | $0.252_{(0.002)}$ | 10.31 |
| **All-Atom Glycan Encoders** | | | | | | | | | | | | |
| All-Atom RGCN | $0.637_{(0.001)}$ | $0.624_{(0.007)}$ | $0.293_{(0.014)}$ | $0.156_{(0.028)}$ | $0.112_{(0.023)}$ | $0.096_{(0.006)}$ | $0.063_{(0.007)}$ | $0.035_{(0.005)}$ | $0.520_{(0.017)}$ | $0.928_{(0.017)}$ | $0.215_{(0.003)}$ | 19.88 |
| Graphormer | *$0.640_{(0.006)}$* | $0.468_{(0.054)}$ | $0.249_{(0.041)}$ | $0.201_{(0.013)}$ | $0.142_{(0.019)}$ | $0.112_{(0.009)}$ | $0.077_{(0.006)}$ | $0.054_{(0.044)}$ | $0.637_{(0.062)}$ | $0.856_{(0.009)}$ | $0.211_{(0.027)}$ | 22.91 |
| GraphGPS | $0.477_{(0.002)}$ | $0.511_{(0.040)}$ | $0.314_{(0.022)}$ | $0.261_{(0.051)}$ | $0.153_{(0.018)}$ | $0.134_{(0.008)}$ | $0.105_{(0.006)}$ | $0.065_{(0.017)}$ | $0.637_{(0.075)}$ | $0.883_{(0.032)}$ | $0.247_{(0.016)}$ | 20.38 |
| Uni-Mol+ | $0.639_{(0.004)}$ | $0.446_{(0.034)}$ | $0.227_{(0.023)}$ | $0.174_{(0.019)}$ | $0.128_{(0.020)}$ | $0.109_{(0.017)}$ | $0.077_{(0.012)}$ | $0.056_{(0.003)}$ | $0.789_{(0.099)}$ | $0.885_{(0.045)}$ | $0.241_{(0.007)}$ | 16.56 |
| GlycanAA-SP | $0.589_{(0.073)}$ | $0.635_{(0.078)}$ | $0.444_{(0.019)}$ | $0.395_{(0.009)}$ | *$0.270_{(0.006)}$* | $0.205_{(0.005)}$ | $0.176_{(0.015)}$ | $0.154_{(0.009)}$ | $0.755_{(0.010)}$ | $0.946_{(0.017)}$ | $0.241_{(0.003)}$ | 11.22 |
| GlycanAA-AN | $0.609_{(0.028)}$ | *$0.685_{(0.001)}$* | $0.453_{(0.037)}$ | *$0.427_{(0.027)}$* | $0.270_{(0.009)}$ | $0.199_{(0.012)}$ | $0.179_{(0.007)}$ | *$0.155_{(0.003)}$* | $0.765_{(0.024)}$ | $0.947_{(0.025)}$ | $0.241_{(0.004)}$ | 10.44 |
| GlycanAA | $0.642_{(0.002)}$ | $0.683_{(0.020)}$ | $0.484_{(0.009)}$ | $0.429_{(0.022)}$ | $0.291_{(0.003)}$ | $0.221_{(0.002)}$ | $0.198_{(0.011)}$ | $0.157_{(0.011)}$ | $0.792_{(0.021)}$ | $0.950_{(0.020)}$ | $0.288_{(0.003)}$ | 2.56 |
| **Pre-trained All-Atom Glycan Encoders** | | | | | | | | | | | | |
| VabsNet | $0.607_{(0.004)}$ | $0.622_{(0.022)}$ | $0.363_{(0.006)}$ | $0.261_{(0.023)}$ | $0.175_{(0.015)}$ | $0.125_{(0.003)}$ | $0.104_{(0.005)}$ | $0.068_{(0.006)}$ | $0.742_{(0.040)}$ | $0.903_{(0.015)}$ | $0.160_{(0.008)}$ | 19.03 |
| GlycanAA-Attribute | $0.628_{(0.007)}$ | $0.687_{(0.001)}$ | $0.457_{(0.028)}$ | $0.392_{(0.033)}$ | $0.263_{(0.011)}$ | *$0.208_{(0.004)}$* | *$0.188_{(0.001)}$* | $0.143_{(0.003)}$ | $0.722_{(0.009)}$ | $0.925_{(0.011)}$ | $0.263_{(0.009)}$ | 10.47 |
| GlycanAA-Context | $0.637_{(0.002)}$ | $0.643_{(0.048)}$ | $0.453_{(0.026)}$ | $0.386_{(0.038)}$ | $0.259_{(0.033)}$ | $0.205_{(0.005)}$ | $0.177_{(0.004)}$ | $0.144_{(0.007)}$ | $0.768_{(0.013)}$ | $0.946_{(0.018)}$ | $0.270_{(0.010)}$ | 7.06 |
| PreGlycanAA | **$0.661_{(0.025)}$** | **$0.688_{(0.001)}$** | **$0.502_{(0.018)}$** | **$0.447_{(0.014)}$** | **$0.297_{(0.005)}$** | **$0.233_{(0.010)}$** | **$0.203_{(0.003)}$** | **$0.174_{(0.004)}$** | *$0.850_{(0.044)}$* | **$0.961_{(0.011)}$** | **$0.297_{(0.002)}$** | **1.5** |

2-layer MLP with GELU activation. In protein-glycan interaction prediction, the ESM-1b pre-trained protein language model (Rives et al., 2021) with fixed model parameters is used to extract protein representations. All implementations are based on the PyTorch (Paszke et al., 2019) and TorchDrug (Zhu et al., 2022) libraries.

**Pre-training setups**: The PreGlycanAA model is pre-trained with an Adam optimizer (learning rate: $5 \times 10^{-4}$, weight decay: $1 \times 10^{-3}$, batch size: 256) for 50 epochs on the curated pre-training dataset (Section 4.1). We set the atom mask ratio $\rho_a$ and the monosaccharide mask ratio $\rho_m$ as 0.45 and 0.15, and the sensitivities of these two parameters are analyzed in Section 5.3. We provide the accuracy and perplexity curves of pre-training in Appendix A.1. The pre-training is conducted on a local server with 200 CPU cores and 10 NVIDIA GeForce RTX 4090 GPUs (24GB).

**Downstream training setups**: Following the standard of GlycanML benchmark, we conduct all experiments on seeds 0, 1 and 2 and report the mean and standard deviation of results. For GlycanAA, we train it with an Adam optimizer (learning rate: $5 \times 10^{-4}$, weight decay: $1 \times 10^{-3}$) for 50 epochs with batch size 256 on taxonomy, immunogenicity

and glycosylation type prediction and for 10 epochs with batch size 32 on interaction prediction. For fine-tuning PreGlycanAA on downstream tasks, we keep other settings the same as GlycanAA except that the learning rate of the encoder part is set as one tenth of that of the following task-specific MLP predictor (*i.e.*, encoder learning rate: $5 \times 10^{-5}$, predictor learning rate: $5 \times 10^{-4}$). For model selection, we perform validation after each training epoch, and the checkpoint with the best validation performance is chosen for test. All downstream experiments are conducted on a local server with 100 CPU cores and 4 NVIDIA GeForce RTX 4090 GPUs (24GB).

## 5.2. Benchmark Results on GlycanML

**Evaluation metrics**: As in the original benchmark, we use Macro-F1 score as the metric for taxonomy and glycosylation type prediction, AUPRC as the metric for immunogenicity prediction, Spearman's $\rho$ as the metric for interaction prediction, and weighted mean rank as the metric for a model's comprehensive performance. Weighted mean rank computes the weighted average of a model's ranks over all tasks, where each taxonomy prediction task

weighs $1/8$ and each of the other three tasks weighs 1, so as to balance between different types of tasks.

**Baselines**: We compare our models with the baselines studied in the GlycanML benchmark (Xu et al., 2024), including four monosaccharide-level glycan sequence encoders (*i.e.*, LSTM (Hochreiter & Schmidhuber, 1997), ResNet (He et al., 2016), Transformer (Vaswani et al., 2017) and Shallow CNN (Shanehsazzadeh et al., 2020)), nine monosaccharide-level glycan graph encoders (GCN (Kipf & Welling, 2017), GAT (Veličković et al., 2017), MPNN (Gilmer et al., 2017), CompGCN (Vashishth et al., 2019), GIN (Xu et al., 2018), RGCN (Schlichtkrull et al., 2018), GearNet (Zhang et al., 2023b), GearNet-Edge (Zhang et al., 2023b) and ProNet (Wang et al., 2023a)), four state-of-the-art all-atom molecular encoders (*i.e.*, Graphormer (Ying et al., 2021), GraphGPS (Rampášek et al., 2022), Uni-Mol+ (Lu et al., 2024) and VabsNet (Zhuang et al., 2024)). Given the strong performance of RGCN on modeling monosaccharide-level glycan graphs as shown in Xu et al. (2024), we additionally evaluate it on modeling the all-atom molecular graphs of glycans, namely All-Atom RGCN, and also pre-train it with a similar mask prediction algorithm as PreGlycanAA, namely PreRGCN. The pre-training effectiveness of PreGlycanAA and PreRGCN are compared in Appendix A.2. To study pre-training more in depth, we employ the pre-training methods, attribute masking and context prediction, proposed in Hu et al. (2019) to pre-train GlycanAA, deriving the GlycanAA-Attribute and GlycanAA-Context models to compare with PreGlycanAA.

**Results**: In Table 1, we report the performance of the proposed models and various baselines. Based on these results, we highlight the findings below:

- **The superiority of GlycanAA over existing glycan encoders illustrates the benefits of all-atom glycan modeling.** GlycanAA outperforms the best baseline result on 10 out of 11 tasks and also surpasses all baselines in terms of weighted mean rank. It is worth noticing that, in terms of weighted mean rank, GlycanAA also outperforms the PreRGCN model pre-trained with a similar approach as PreGlycanAA. Therefore, it is beneficial to utilize atomic-level information in addition to monosaccharide-level information, and the advantage of GlycanAA derives from well leveraging both kinds of information. Also, the superiority of GlycanAA over PreRGCN illustrates the importance of hierarchical structures to our pre-training method.

- **The performance gains of PreGlycanAA over GlycanAA demonstrate the effectiveness of the proposed pre-training method.** PreGlycanAA outperforms GlycanAA on all 11 tasks and ranks first among all models in terms of weighted mean rank. Given the same model architecture between PreGlycanAA and GlycanAA, we confirm that the proposed multi-scale pre-training method can enhance the model capability. The obvious advantage of PreGlycanAA over GlycanAA-Attribute and GlycanAA-Context demonstrates that the proposed multi-scale mask prediction method is well-suited to self-supervised glycan representation learning.

- **Directly applying performant small molecule encoders or monosaccharide-level glycan encoders to all-atom glycan modeling is unpromising.** Graphormer, GraphGPS and Uni-Mol+ have been shown to be effective in modeling small molecules with tens of atoms (Shi et al., 2022). However, benchmark results show that they do not perform well when modeling all-atom molecular graphs of glycans with hundreds of atoms. Similarly, compared to the well-performing monosaccharide-level RGCN, the performance of All-Atom RGCN is unsatisfactory. Thus, dedicated designs for all-atom glycan modeling are highly demanded.

### 5.3. Ablation Studies

**Effect of hierarchical message passing**: To study the necessity of hierarchical message passing, we substitute it with a single message passing in each message passing block of GlycanAA, where the single message passing is also implemented as relational graph convolution (Equation (1)). We name this model variant as GlycanAA-SP. By comparing GlycanAA and GlycanAA-SP in Table 1, we can observe the obvious advantages of GlycanAA, where it achieves a better result on all 11 tasks and also on weighted mean rank. These results show the benefit of passing messages hierarchically on the proposed all-atom glycan graph.

**Effect of monosaccharide-wise readout**: In GlycanAA, we by default use monosaccharide-wise readout. Here, we compare this scheme with all-node readout, where mean and max pooling are performed over all atom and monosaccharide nodes. The model variant with all-node readout is named as GlycanAA-AN. According to Table 1, GlycanAA outperforms GlycanAA-AN on 10 out of 11 tasks and also on weighted mean rank. Therefore, monosaccharide-wise readout is a better readout scheme, in which only useful atomic information is retained, leading to more discriminative glycan representations and thus better performance.

**Sensitivity of PreGlycanAA to mask ratio**: In this experiment, we analyze how different atom and monosaccharide mask ratios affect the performance of PreGlycanAA on downstream tasks. Specifically, we uniformly select atom and monosaccharide mask ratios between 0 and 1 with the interval of 0.15 and combine them into 36 pairs: $(\rho_a, \rho_m) \in \{0.15, 0.3, 0.45, 0.6, 0.75, 0.9\} \times \{0.15, 0.3, 0.45, 0.6, 0.75, 0.9\}$. We pre-train a model under each mask ratio pair and evaluate its performance on

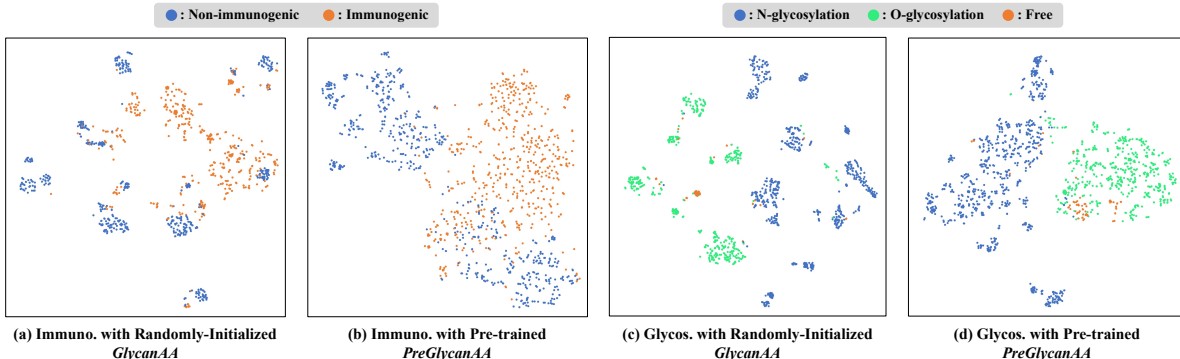

**(a) Immuno. with Randomly-Initialized** *GlycanAA*  **(b) Immuno. with Pre-trained** *PreGlycanAA*  **(c) Glycos. with Randomly-Initialized** *GlycanAA*  **(d) Glycos. with Pre-trained** *PreGlycanAA*

Figure 3: Visualization of glycan representations extracted by GlycanAA and PreGlycanAA on downstream task datasets. *Abbr.*, Immuno.: Immunogenicity; Glycos.: Glycosylation.

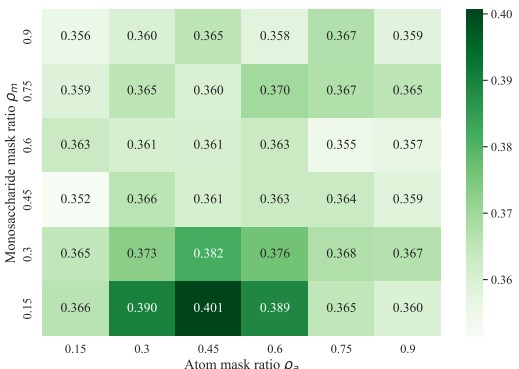

Figure 4: Average Macro-F1 score of PreGlycanAA on eight taxonomy prediction tasks under different atom and monosaccharide mask ratios.

eight glycan taxonomy prediction tasks. In Figure 4, we visualize the average Macro-F1 score on eight tasks for all models. The pre-trained model achieves prominent performance when $\rho_a$ is around 0.45 and $\rho_m$ is around 0.15. Under such settings, a suitable balance is achieved between masked and observed information in a glycan, and thus the model can be effectively pre-trained.

### 5.4. Computational Efficiency Study

To evaluate the additional computational cost brought by all-atom glycan modeling compared to monosaccharide-level modeling, we study the computational efficiency of GlycanAA against a well-performing monosaccharide-level glycan encoder, RGCN. Specifically, we evaluate their training and inference speed in terms of throughput (*i.e.*, the number of samples processed in one second) and their training and inference memory cost in terms of Mebibyte (MiB). Evaluation details are stated in Appendix A.4.

In Table 2, we present the efficiency comparisons between RGCN and GlycanAA. For training/inference speed, GlycanAA is about 22% slower than RGCN, and, for training/inference memory cost, GlycanAA consumes about 19% more memory than RGCN. Such a moderate extra

Table 2: Efficiency comparison between RGCN and GlycanAA on the taxonomy prediction dataset.

| Model | Training speed (#samples/s) | Inference speed (#samples/s) | Training memory cost (MiB) | Inference memory cost (MiB) |
|---|---|---|---|---|
| RGCN | 885.7 | 1486.9 | 6911.6 | 3563.5 |
| GlycanAA | 679.8 | 1158.6 | 8213.9 | 4251.2 |

cost brings the superior performance of GlycanAA over RGCN on all 11 benchmark tasks and also on the weighted mean rank (shown in Table 1), illustrating the "worth" of modeling glycans on the all-atom level.

### 5.5. Visualization

To intuitively study the effect of pre-training, we visualize the glycan representations extracted by the GlycanAA with random weights and the PreGlycanAA with pre-trained weights, respectively. We use t-SNE (Van der Maaten & Hinton, 2008) for visualization. The results on the datasets of immunogenicity and glycosylation type prediction are shown in Figure 3, and more results are in Appendix A.3.

In Figure 3, after pre-training, the model can more effectively separate the samples of different classes and gather the samples of the same class together, leading to smoother decision boundaries. This effect leads to better generalization performance of PreGlycanAA over GlycanAA on immunogenicity and glycosylation type prediction. These visualization results provide a way to interpret how pre-training benefits downstream glycan understanding tasks.

## 6. Conclusions and Future Work

We propose the GlycanAA model to encode heterogeneous all-atom glycan graphs with hierarchical message passing. GlycanAA is further pre-trained on a set of high-quality unlabeled glycans through multi-scale mask prediction, deriving the PreGlycanAA model. On the GlycanML benchmark, we illustrate the superiority of GlycanAA and PreGlycanAA over existing glycan encoders.

In the future, we will focus on boosting real-world glycan-related applications with the proposed models. For example, we will study how vaccine design and cancer research can be promoted by all-atom glycan machine learning models.

## Impact Statement

This work aims to build all-atom glycan machine learning models and use the models to well tackle various glycan understanding tasks, including glycan taxonomy prediction, glycan immunogenicity prediction, glycosylation type prediction and protein-glycan interaction prediction. The proposed models can potentially promote real-world glycan-related applications such as vaccine design (Kaplonek et al., 2018) and cancer research (Taniguchi & Kizuka, 2015).

However, we should not ignore the potential risks brought by glycan machine learning models, *e.g.*, designing vaccines with severe adverse reactions. To mitigate such risks, our future work will encourage the responsible usage of the proposed models for real-world problems.

## Acknowledgments

This work is supported by the National Key R&D Program of China (2024YFA1014003), National Natural Science Foundation of China (92470121, 62402016), CAAI-Ant Group Research Fund, and High-performance Computing Platform of Peking University.

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

# A. Appendix

## A.1. Accuracy and Perplexity Curves during Pre-training

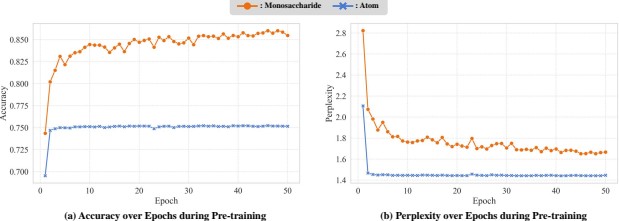

(a) Accuracy over Epochs during Pre-training  (b) Perplexity over Epochs during Pre-training

Figure 5: The accuracy and perplexity curves during the pre-training phase of PreGlycanAA.

In this appendix, we present the accuracy and perplexity curves that are obtained during the pre-training phase of PreGlycanAA. These curves provide valuable insights into the learning dynamics and the effectiveness of the proposed pre-training method.

**Accuracy curve**: The accuracy curves in Figure 5(a) illustrate the model's ability to recover masked atoms and monosaccharides correctly along the pre-training process. The initial steep incline suggests rapid learning in the early stage, followed by a gradual approach towards an asymptote, signifying the model's convergence. We can observe the slower convergence of the monosaccharide recovery accuracy compared to the atom recovery accuracy, indicating that the masked monosaccharide prediction task is harder to learn.

**Perplexity curve**: Perplexity is a measurement of how well a probability distribution predicts a sample, often used in the context of language modeling (Devlin, 2018). A lower perplexity indicates that the model is more confident at recovering masked elements to their true values. The perplexity curves in Figure 5(b) reflect the reduction of model's uncertainty as pre-training proceeds. Similar to accuracy curves, the convergence of the monosaccharide recovery perplexity is slower than that of the atom recovery perplexity, again indicating the higher difficulty of the masked monosaccharide prediction task.

## A.2. Effect of Model Architecture on Pre-training

In this study, we investigate the effect of model capacity on solving the pre-training task. We select two typical models: (1) the GlycanAA model that models glycans on both monosaccharide and atom levels, and (2) the RGCN model that only performs monosaccharide-level modeling. In Figure 6, we present the accuracy and cross entropy loss curves of pre-training for these two models. According to the results, compared to RGCN, GlycanAA performs clearly better in pre-training with higher accuracy and lower cross entropy loss, thanks to its higher model capacity.

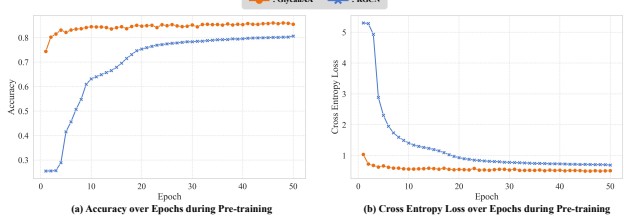

(a) Accuracy over Epochs during Pre-training  (b) Cross Entropy Loss over Epochs during Pre-training

Figure 6: The accuracy and cross entropy loss curves of masked monosaccharide prediction during pre-training GlycanAA and RGCN.

By checking the benchmark results in Table 1, we can observe that the pre-trained GlycanAA (*i.e.*, PreGlycanAA) achieves clearly more performance gains on downstream tasks after pre-training, compared to the pre-trained RGCN (*i.e.*, PreRGCN). This correlation between higher model capacity, higher pre-training performance and more performance gains on downstream tasks is also reported in other domains (Devlin, 2018; He et al., 2020; Hu et al., 2019).

## A.3. Additional Visualization of Glycan Representations

In Figure 7, we present the glycan representations extracted by GlycanAA and PreGlycanAA on the datasets of eight glycan taxonomy prediction tasks, where GlycanAA is randomly initialized and PreGlycanAA is pre-trained. We employ the t-SNE algorithm (Van der Maaten & Hinton, 2008) for dimensionality reduction.

According to these results, we can observe the better clustering behavior of PreGlycanAA, where it more effectively separates the samples of different classes and gathers the samples of the same class together. This phenomenon is more visually significant on the tasks with fewer classes, *e.g.*, domain and kingdom prediction tasks. The better clustering behavior of PreGlycanAA leads to its superior performance over GlycanAA on all 8 taxonomy prediction tasks, as shown in Table 1.

## A.4. Evaluation Details of Computational Efficiency Study

We evaluate the training and inference speed of GlycanAA and RGCN in terms of throughput (*i.e.*, the number of samples processed in one second) and their training and inference memory cost in terms of Mebibyte (MiB). The evaluation is performed on the dataset of glycan taxonomy prediction for its good coverage of different kinds of glycans (#training/validation/test samples: 11,010/1,280/919, average #monosaccharides per glycan: 6.39, minimum #monosaccharides per glycan: 2, maximum #monosaccharides per glycan: 43). All experiments are conducted on a machine with 32 CPU cores and 1 NVIDIA GeForce RTX 4090 GPU (24GB), and the batch size is set as 256.

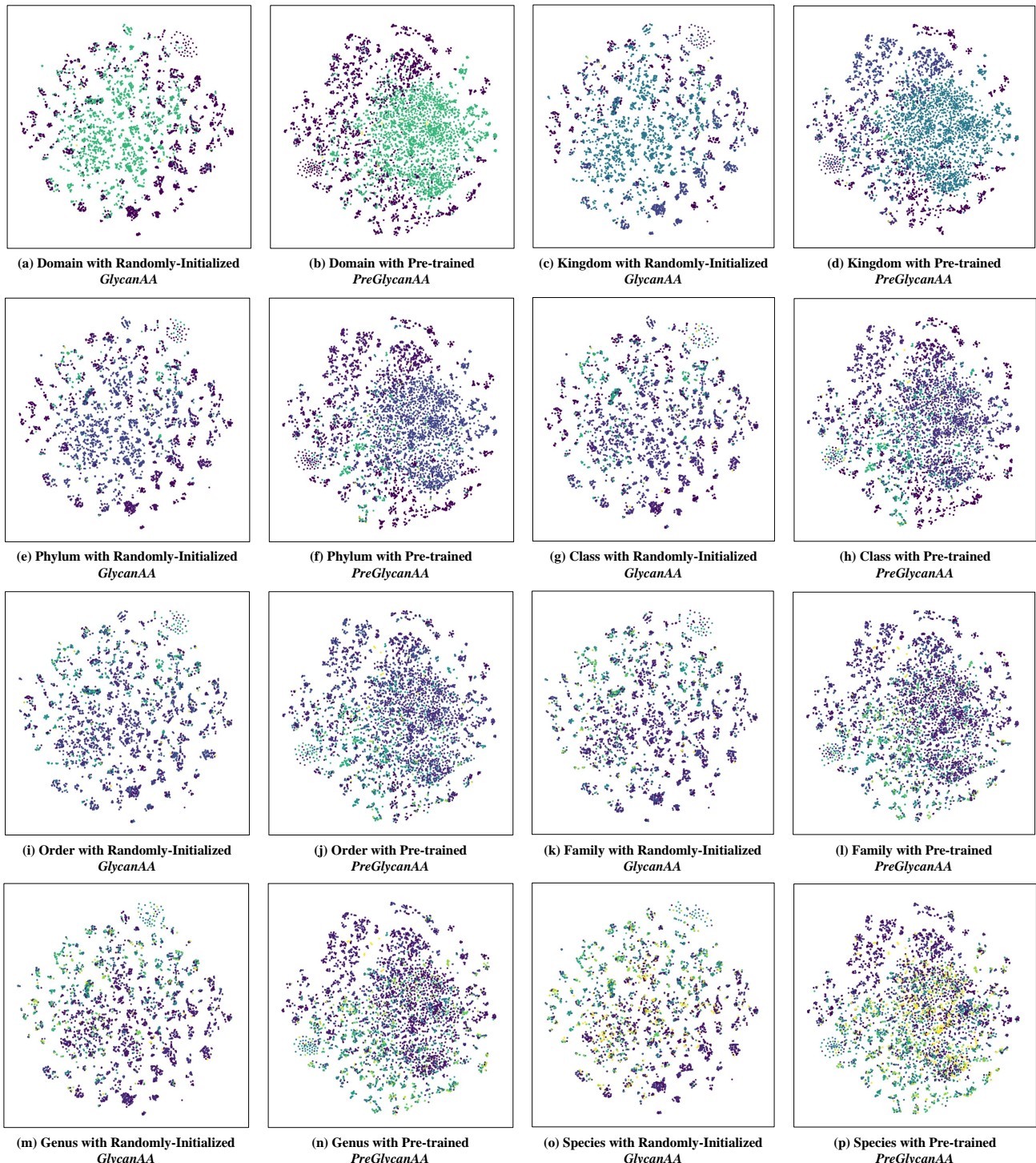

Figure 7: Visualization of glycan representations extracted by GlycanAA and PreGlycanAA on taxonomy prediction tasks. We use different colors to indicate the glycans of different classes, and the color-class correspondence is omitted for concision (many tasks own hundreds of classes).

