# OpenReview forum: "Modeling All-Atom Glycan Structures via Hierarchical Message Passing and Multi-Scale Pre-training"
_ICML.cc/2025/Conference — ICML 2025 poster_

### Official Review · Reviewer_rHxW · 2025-03-12

**Overall Recommendation:** 2

**Summary:**

The paper introduces a hierarchical GNN for all-atom glycan modeling supported by a multi-scale pre-training strategy.

**Claims And Evidence:**

The paper is well-written and easy to understand.

**Essential References Not Discussed:**

I did not recognize it.

**Experimental Designs Or Analyses:**

I notice that when a similar pre-training strategy is applied to RGCN (as PreRGCN), its performance actually gets worse. Could you clarify why this pre-training fails to help—or even harms—RGCN, while it improves the proposed model?

**Methods And Evaluation Criteria:**

Yes

**Other Comments Or Suggestions:**

I’m also concerned about the efficiency trade-off. The authors note that their all-atom approach is about 20% slower than the baseline methods, so I’m wondering how this slowdown affects broader usability or scalability for large-scale glycan datasets.

**Other Strengths And Weaknesses:**

Although glycans have unique features and the field is new, I am still worried about the novelty. The paper’s use of hierarchical message passing and multi-scale pre-training has been done in graph representation learning. I hope the authors can provide a clearer explanation of what is truly new beyond applying existing GNN methods to the glycan domain.

**Questions For Authors:**

Please refer to previous sections.

**Relation To Broader Scientific Literature:**

The paper applies techniques in graph learning like hierarchical message passing and multi-scale pre-training in the field of glycans.

**Theoretical Claims:**

There are no theoretical claims in the paper.

---

> ### Author Rebuttal · Authors · 2025-03-31
>
> Thanks for your valuable comments! We respond to your concerns as below:
>
> >**Q1: Why is the pre-training method less helpful to RGCN than to GlycanAA?**
>
> We deem that **the less benefit of pre-training to RGCN mainly owes to its lower model capacity**. Compared to GlycanAA that models both monosaccharide-level and atomic-level glycan structures, RGCN only models monosaccharide-level structures and is thus with lower capacity. As shown in the figure of [this URL](https://anonymous.4open.science/r/GlycanAA-73E6/pretrain_caption.png), the lower capacity of RGCN leads to its inferior pre-training performance (i.e., lower accuracy and higher cross entropy loss on masked monosaccharide prediction) against GlycanAA. These shortages of RGCN (i.e., lower model capacity and inferior pre-training performance) makes it benefit less on downstream tasks after pre-training, which is consistent with previous findings in other domains [a,b,c].
>
> &emsp;
>
> [a] A simple framework for contrastive learning of visual representations. ICML, 2020.
>
> [b] Bert: Pre-training of deep bidirectional transformers for language understanding. NAACL, 2019.
>
> [c] Strategies for pre-training graph neural networks. ICLR, 2020.
>
> >**Q2: What is truly new beyond applying existing GNN methods to the glycan domain?**
>
> We argue that the proposed methods are **clearly motivated** and **carefully designed** to **handle the complexity of modeling atomic-level glycan structures**, instead of simply applying existing techniques to the glycan domain.
>
> (1) For the model part, *our model design is inspired by the fact that the backbone structure of a glycan mainly determines its biological properties, and the atomic-level structures of individual monosaccharides provide auxiliary information.* Following this principle, **we perform three steps of message passing to progressively enhance global backbone structural features with local atomic structural features**, and the enhanced backbone features are finally readout for glycan representation.
>
> (2) For the algorithm part, *our pre-training algorithm is motivated by the fact that understanding the interactions between monosaccharides and their corresponding atoms is important to effective hierarchical learning, while only performing supervised learning on downstream tasks cannot guarantee acquiring such interactions.* Therefore, in our pre-training algorithm, **we facilitate the model to learn the interactions between monosaccharides and atoms.** Specifically, we first perform an interactive masking process where each selected monosaccharide is masked along with its corresponding atoms, and, on such a masked glycan, the model learns to recover masked monosaccharides with the hints brought by the recovery of some of their atoms.
>
> >**Q3: How the extra cost of GlycanAA affects its usage on large-scale glycan datasets?**
>
> Table A: Efficiency and effectiveness comparison between RGCN and GlycanAA on GlycanDomain-60K.
>
> |Model|Total processing time (s)|Macro-F1|
> |:----:|:----:|:----:|
> |RGCN|436.70|0.282|
> |GlycanAA|489.98|0.425|
>
> &emsp;
>
> This question is great. To study the usability of GlycanAA on large-scale glycan datasets, we construct the GlycanDomain-60K dataset. We first collect all existing glycans deposited in the GlyTouCan database whose structures are complete (GlyTouCan is a regularly updated glycan database containing all discovered glycans), summing up to 60,152 samples. We then annotate each of them with a domain label (Eukarya, Virus, Bacteria or Archaea) based on their nearest neighbor in the domain classification dataset of the GlycanML benchmark, where the nearest neighbor is determined by a motif matching algorithm depicted in [d]. We name this large-scale glycan dataset with domain annotations as GlycanDomain-60K.
>
> On this dataset, we compare the efficiency and effectiveness of RGCN (the most competitive monosaccharide-level baseline) and GlycanAA (our all-atom-level encoder). In specific, we respectively use the glycan domain classifier with RGCN and GlycanAA backbones to predict the domain labels of all glycans in GlycanDomain-60K, where both models are trained on the domain classification task of GlycanML. In Table A, we report the total processing time and the Macro-F1 score of predictions for these two models, where the test is done on a machine with 48 CPU cores and 1 NVIDIA GeForce RTX
> 4090 GPU under the batch size 256. According to the results, *GlycanAA achieves a 50% higher Macro-F1 score than RGCN with less than 1 minute more processing time.* Therefore, **GlycanAA is applicable in processing large-scale glycan datasets, which achieves outstanding performance with little efficiency trade-off.**
>
> In the revised paper, we will add this analysis to better illustrate the usability and scalability of GlycanAA on large-scale glycan datasets.
>
> &emsp;
>
> [d] A motif-based analysis of glycan array data to determine the specificities of glycan-binding proteins. Glycobiology, 2010.

---

> > ### Comment · Reviewer_rHxW · 2025-04-08
> >
> > I thank the authors for their rebuttal. Some of my concerns have been addressed. However, I remain unconvinced for the Q1.
> >
> > Your response indicates that RGCN—pretrained only at the monosaccharide level—suffers in capacity. However, [1] suggests that motif-level pretraining can boost downstream performance rather than limit it.
> >
> > [1] Zhang, Zaixi, et al. "Motif-based graph self-supervised learning for molecular property prediction." Advances in Neural Information Processing Systems 34 (2021): 15870-15882.

---

> > > ### Author Response · Authors · 2025-04-09
> > >
> > > Dear reviewer,
> > >
> > > Thanks for your feedback. We would like to clarify that **the PreRGCN model studied in this work is pre-trained using a masked motif (monosaccharide) prediction task, which is in principle different from the autoregressive motif generation task used for pre-training in [1].** Analogy to the NLP domain, the BERT models based on masked language modeling (similar to our pre-training method) and the GPT models based on autoregressive generation (similar to the pre-training method of [1]) are two different kinds of models.
> > >
> > > In this work, we show that, **for the glycan domain, mask-modeling-like pre-training benefits atomic-level modeling (PreGlycanAA) more than monosaccharide-level modeling (PreRGCN), while monosaccharide-level modeling is stilled benefited (PreRGCN outperforms the non-pretrained RGCN on 9 out of 11 downstream tasks).** Of course, autoregressive-generation-like pre-training is a promising way to boost glycan modeling both on atomic level and on monosaccharide level. We leave this exploration as our important future work.
> > >
> > > &emsp;
> > >
> > > Best,
> > >
> > > Authors
> > >
> > > &emsp;
> > >
> > > [1] Zhang, Zaixi, et al. "Motif-based graph self-supervised learning for molecular property prediction." Advances in Neural Information Processing Systems 34 (2021): 15870-15882.

---

### Official Review · Reviewer_YAu1 · 2025-03-13

**Overall Recommendation:** 3

**Summary:**

The paper introduces GlycanAA, a novel framework for All-Atom Glycan Modeling using hierarchical message passing and self-supervised pretraining. It models glycans as heterogeneous graphs where atom nodes represent local structures and monosaccharide nodes represent the global backbone structure. GlycanAA employs Hierarchical Message Passing to capture atomic-level interactions and glycosidic bonds in a unified framework. The pre-trained model, PreGlycanAA, uses Multi-Scale Mask Prediction for self-supervised learning, enhancing representation power.

**Claims And Evidence:**

Yes. The claims are supported by extensive benchmarking on the GlycanML dataset as shown Table 1.

**Essential References Not Discussed:**

No.

**Experimental Designs Or Analyses:**

Yes. The experimental design is valid, with clear definitions of training, validation, and testing protocols.The ablation studies demonstrate the importance of hierarchical message passing and the superiority of monosaccharide-wise readout.

**Methods And Evaluation Criteria:**

Yes. The GlycanML benchmark is suitable for evaluating glycan properties.

**Other Comments Or Suggestions:**

No.

**Other Strengths And Weaknesses:**

Strengths:
1. This paper introduces a new way of modeling glycans as heterogeneous graphs.
2. The model effectively leverages both local atomic information and global structural information.
3. The multi-scale mask prediction effectively captures glycan dependencies.

Weaknesses:
1. Only uses glycosidic bonds for modeling backbone structures, potentially missing other structural features. Have you considered other graph construction methods? How to get the graph edges, based on the distances?
2. It is better to add a figure to present the Glycans, it's importance.
3. Overhead of All-Atom Modeling: Computationally more expensive than monosaccharide-level modeling. Though the  efficiency comparison is presented in Table 2, it is better to compare it with baselines. In table 1, are the results tested by yourselves? if yes, it is convenient to provide the comparisons of training and testing time.

**Questions For Authors:**

1. Can the model handle more complex glycans with diverse glycosidic linkages?
2. What happens if structural features (e.g., torsion angles) are included in the model?

**Relation To Broader Scientific Literature:**

Builds on previous monosaccharide-level GNNs and small-molecule encoders.
Integrates self-supervised learning, commonly used for proteins and small molecules, into glycan modeling.

**Theoretical Claims:**

No. This paper is not a theoretical paper.

---

> ### Author Rebuttal · Authors · 2025-03-31
>
> Thanks for your valuable comments and constructive suggestions! We respond to your questions as below:
>
> >**Q1: Can the model handle more complex glycans with diverse glycosidic linkages?**
>
> We announce that **the proposed GlycanAA model can handle any glycan no matter how complex its structure is**. Basically, for a given glycan, GlycanAA extracts each of its monosaccharide as a node in the backbone-level graph, and, for each monosaccharide, its fine-grained atomic-level structure is further modeled by an atomic graph; GlycanAA constructs relational edges between different monosaccharides to capture glycosidic linkages, where **it use 84 types of relations to model all possible glycosidic bonds that connect atoms at different sites with different stereochemical configurations**.
>
> The glycans with complex structures can be well modeled in this way. For example, starch is a complex kind of glycans composed of hundreds of glucoses (monosaccharide units) and thousands of $\alpha$1-4 and $\alpha$1-6 glycosidic bonds. By using the proposed backbone- and atomic-level graph modeling and multi-relational glycosidic bond modeling approaches, GlycanAA can well capture (1) the local structure within each glucose unit and (2) the global structures of starch formed by $\alpha$1-4 glycosidic bonds for its linear parts and $\alpha$1-6 glycosidic bonds for its branching parts.
>
> >**Q2: What happens if structural features are included in the model?**
>
> Table A: Performance comparison between GlycanAA and GlycanAA-torsion on taxonomy prediction tasks. The Macro-F1 score for each task and the mean Macro-F1 score over all tasks are reported.
>
> |Model|Domain|Kingdom|Phylum|Class|Order|Family|Genus|Species|Mean Macro-F1|
> |:----:|:----:|:----:|:----:|:----:|:----:|:----:|:----:|:----:|:----:|
> |GlycanAA|0.642|0.683|0.484|0.429|0.291|**0.221**|**0.198**|**0.157**|0.388|
> |GlycanAA-torsion|**0.651**|**0.687**|**0.486**|**0.437**|**0.302**|0.215|0.190|0.154|**0.390**|
>
> &emsp;
>
> This question is great. First, we claim that **GlycanAA owns a generic model framework which can easily incorporate various structural features**. Here, we take the featurization of torsion angles of glycosidic bonds as an example. Specifically, by using the Carbohydrate Builder of GLYCAM, we get the 3D conformation of each glycan data in the dataset for taxonomy prediction. For each glycosidic bond in a glycan structure, it defines two torsion angles $\phi$ and $\psi$, as shown in the figure of [this URL](https://anonymous.4open.science/r/GlycanAA-73E6/torsion_caption.png). To include these two torsion angles in GlycanAA, we respectively compute their sine and cosine values and map the resulting four values to the hidden space for message passing. We name this model variant as *GlycanAA-torsion*.
>
> In Table A, we compare the performance of GlycanAA and GlycanAA-torsion on eight taxonomy prediction tasks. According to the results, GlycanAA-torsion outperforms GlycanAA on 5 tasks with at most 210 taxonomy categories, while GlycanAA-torsion is inferior on 3 tasks with at least 415 taxonomy categories. These results demonstrate that including torsion angle features can enhance the model's ability to fit the data, but it can also make the model more prone to overfitting, especially in complex tasks.
>
> **We will include this study in the revised paper to inspire more future work on glycan structure modeling**, e.g., constructing distance-induced glycan graphs based on the vicinity of atoms and monosaccharides in 3D glycan conformations. **We will also include the figure presenting glycosidic torsion angles in the revision for better understanding of glycan structures.**
>
> >**Q3: The selection of baseline for efficiency study.**
>
> We clarify that, in the efficiency study, we select the best-performing baseline (without pre-training), i.e., RGCN, for comparison with GlycanAA. Comparing to RGCN, GlycanAA performs better on all 11 benchmark tasks with a moderate 20% more computational cost, demonstrating that **GlycanAA achieves remarkable performance gains against the most competitive baseline under acceptable efficiency trade-offs**.

---

> > ### Comment · Reviewer_YAu1 · 2025-04-06
> >
> > Thanks for the rebuttal. Most of my questions are tackled.

---

### Official Review · Reviewer_DPXx · 2025-03-22

**Overall Recommendation:** 4

**Summary:**

This paper proposes a hierarchical graph model for atom-level glycan modeling. It employs self-supervised learning to enhance the model's capability. The self-supervised learning framework uses multi-scale mask prediction as its task. Subsequently, the pre-trained model is utilized for downstream tasks. The hierarchical graph network effectively models atom-level glycan structures. Through this pre-training and fine-tuning process, the proposed model surpasses previous state-of-the-art methods.

**Claims And Evidence:**

The submission includes claims that are supported by clear and convincing evidence. Experimental results and analyses validate these claims, demonstrating the effectiveness of the proposed hierarchical graph model.

**Essential References Not Discussed:**

The study titled [ProNet'22] employs hierarchical graph networks for protein 3D modeling. This paper compares the proposed method with GearNet, which was originally designed for protein 3D structures. Therefore, the authors should consider discussing [ProNet'22] and potentially using it as a baseline for further comparison.

ProNet'22: Learning Hierarchical Protein Representations via Complete 3D Graph Networks

**Experimental Designs Or Analyses:**

The experimental design makes sense. However, an ablation study focusing on the proposed hierarchical graph during the pre-training stage could further demonstrate the paper's effectiveness. For instance, comparing the loss curves with and without the atom-level graph would provide valuable insights.

**Methods And Evaluation Criteria:**

The proposed method and evaluation criteria make sense for the application.

**Other Comments Or Suggestions:**

No.

**Other Strengths And Weaknesses:**

No.

**Questions For Authors:**

See the "Experimental Designs Or Analyses" and "Essential References Not Discussed".

**Relation To Broader Scientific Literature:**

No

**Theoretical Claims:**

This paper does not propose theoretical claims.

---

> ### Author Rebuttal · Authors · 2025-03-31
>
> Appreciate your insightful comments and golden suggestions! We respond as below:
>
> >**Q1: An ablation study focusing on the proposed hierarchical graph during the pre-training stage is recommended.**
>
> This suggestion is great. By removing the atom-level modeling part, the obtained model variant of GlycanAA essentially performs relational message passing among monosaccharides, which is basically RGCN. Therefore, we compare the pre-training performance of GlycanAA and RGCN by using the proposed mask prediction algorithm, where the monosaccharide mask ratio is set as 0.3 for both models. In the figure of [this URL](https://anonymous.4open.science/r/GlycanAA-73E6/pretrain_caption.png), we present the accuracy and cross entropy loss curves of pre-training for these two models. According to the results, compared to RGCN, GlycanAA performs clearly better in pre-training with higher accuracy and lower cross entropy loss, thanks to its higher model capacity (i.e., modeling both monosaccharide-level and atom-level glycan structures). By checking the benchmark results in the Table 1 of paper submission, we can observe that the pre-trained GlycanAA (i.e., PreGlycanAA in the table) achieves clearly more performance gains on downstream tasks after pre-training, compared to the pre-trained RGCN (i.e., PreRGCN in the table). This correlation between higher model capacity, higher pre-training performance and more performance gains on downstream tasks is also reported in other domains [a,b,c].
>
> We will add this study to the revised paper version, so as to give more insights to pre-training glycan representations.
>
> &emsp;
>
> [a] A simple framework for contrastive learning of visual representations. ICML, 2020.
>
> [b] Bert: Pre-training of deep bidirectional transformers for language understanding. NAACL, 2019.
>
> [c] Strategies for pre-training graph neural networks. ICLR, 2020.
>
> > **Q2: ProNet is a related work, which should be discussed and compared with.**
>
> Table A: Performance comparison between ProNet, GearNet, GearNet-Edge and GlycanAA on benchmark tasks. The best and second-best results are denoted by **bold** and *italic*, respectively.
>
> |Model|Domain|Kingdom|Phylum|Class|Order|Family|Genus|Species|Immunogenicity|Glycosylation|Interaction|Weighted Mean Rank|
> |:----:|:----:|:----:|:----:|:----:|:----:|:----:|:----:|:----:|:----:|:----:|:----:|:----:|
> |ProNet|0.627|*0.590*|*0.438*|0.380|0.242|0.192|0.146|0.128|*0.778*|*0.930*|*0.252*|*2.31*|
> |GearNet|0.471|0.577|0.395|*0.389*|0.256|0.189|0.165|0.136|0.740|0.892|0.248|3.81|
> |GearNet-Edge|*0.628*|0.573|0.396|0.384|*0.262*|*0.200*|*0.177*|*0.140*|0.768|0.909|0.250|2.88|
> |GlycanAA|**0.642**|**0.683**|**0.484**|**0.429**|**0.291**|**0.221**|**0.198**|**0.157**|**0.792**|**0.950**|**0.288**|**1.00**|
>
> &emsp;
>
> Thanks for pointing this out. ProNet [d] is a representative model for protein structure modeling, which simultaneously captures the amino-acid-level, backbone-level and all-atom-level structures of a protein. In its implementation, ProNet passes the structural features at higher resolution (i.e., backbone-level and all-atom-level features) to the structural features at lower resolution (i.e., amino-acid-level features), and the graph convolution at lower resolution is biased by the features passed from higher resolution.
>
> To investigate such a modeling approach in the glycan domain, we implement a ProNet for glycan modeling which follows the original architecture with four interaction blocks, and, in each interaction block, atom-level features are passed to monosaccharide-level features to bias the graph convolution operation. In Table A, we compare this ProNet with GearNet, GearNet-Edge and GlycanAA on benchmark tasks. According to the results, ProNet outperforms GearNet and GearNet-Edge in terms of weighted mean rank, while it is inferior to GlycanAA on all benchmark tasks. This result again demonstrates the effectiveness of the proposed hierarchical relational message passing scheme in GlycanAA, which well captures different kinds of dependencies within a glycan.
>
> In the revised paper, we will supplement the above discussion and comparison for the interests of a broader audience.
>
> &emsp;
>
> [d] Learning Hierarchical Protein Representations via Complete 3D Graph Networks. ICLR, 2023.

---

### Decision · Program_Chairs · 2025-05-01

**Decision:**

Accept (poster)

**Comment:**

This submission focuses on a hierarchical graph model for atom-level glycan modeling. It employs self-supervised learning to enhance the model's capability. The self-supervised learning framework uses multi-scale mask prediction as its task. Most concerns are about baseline comparisons, which we believe are well addressed during the rebuttal. Overall, we think it is a good paper and recommend acceptance.